# A Rare Case of Severe Diarrhea: Gastrocolic Fistula Caused by Migration of Percutaneous Endoscopic Gastrostomy Tube

**DOI:** 10.3390/healthcare11091263

**Published:** 2023-04-28

**Authors:** Maria Elena Pugliese, Riccardo Battaglia, Antonio Cerasa, Lucia Francesca Lucca

**Affiliations:** 1Severe Acquired Brain Injury Unit, S’Anna Institute, 88900 Crotone, Italy; 2Institute for Biomedical Research and Innovation (IRIB), National Research Council of Italy (CNR), 98164 Messina, Italy; 3Pharmacotechnology Documentation and Transfer Unit, Preclinical and Translational Pharmacology, Department of Pharmacy, Health and Nutritional Sciences, University of Calabria, 87036 Rende, Italy

**Keywords:** percutaneous endoscopic gastrostomy, PEG, traumatic brain injury, severe diarrhea

## Abstract

Gastrocolic fistula is a rare complication of the percutaneous endoscopic gastrostomy (PEG) placement procedure. This complication occurs due to penetration of interposed colon when a PEG tube is placed into the stomach. It can go unrecognized, becoming evident only when a tube replacement is performed or tube migration occurs. We report a case of severe, intractable diarrhea occurring about one month after the PEG procedure in a patient with severe traumatic brain injury. We present our case and discuss its significance with the aim of raising clinicians’ awareness of this rare condition.

## 1. Introduction

Gastrocolic fistula (GCF) was first described as a complication of gastric cancer [1]. It has since been observed in association with several processes, including malignancy and chronic inflammatory bowel diseases, and has also been reported from iatrogenic causes such as percutaneous gastrostomy tube migration [2]. 

According to the guidelines of the European Society of Clinical Nutrition (ESPEN), percutaneous endoscopic gastrostomy (PEG) tube placement is indicated for patients requiring long-term enteral nutritional support [3]. Over the past decades, it has proven as a safe and effective technique for enteral feeding. Complications associated with PEG are wound infection, gastric perforation, bleeding and colonic injury, peristomal leak, and tube dislodgement [4]. Gastrocoloic or gastrocolocutaneous fistula is a rare complication associated with PEG, with an incidence of 0.5 to 3% [5]. This complication develops from the accidental perforation of the colon interposed between the abdominal and stomach wall during PEG placement. Affected patients may remain asymptomatic for several months. It usually becomes symptomatic as soon as the tube is replaced or a spontaneous tube migration occurs. Common symptoms include abdominal pain, nausea, vomiting, and diarrhea. Patients may also experience weight loss, electrolyte imbalances, and malnutrition [6].

We report a case of a gastrocolocutaneous fistula that presented with severe, intractable diarrhea in a post-traumatic patient. 

## 2. Materials and Methods

### Case Report

A 36 years-old patient was admitted to our intensive rehabilitation unit (IRU) to undergo an intensive rehabilitation program after a severe trauma. He unintentionally tumbles from the second floor. Significant extracerebral bleeding was identified and managed by cranial decompression, with severe neurological impairment (minimally conscious state, Coma Recovery Scale-revised 12 points). The patient additionally reported D11 somatic vertebral fracture with paraplegia (ASIA scale A, level D10). He underwent a button-type PEG placement two weeks before uneventfully. No bloody or tarry stools were noticed after the procedure. His past medical history was unremarkable, except for multiple food intolerances (red meat, milk, dairy products, and most vegetables). An abdominal CT performed at admission time showed the tube properly positioned in the stomach (Figure 1A and Figure 2A–C).

Enteral feeding was well tolerated. Ten days later admission, the onset of watery diarrhea was observed, with more than three deliveries a day. The patient was hemodynamically stable. Physical examination revealed a soft and flat abdomen with normal bowel sounds. At first, the type of industrial feeding and feeding velocity was changed (from pump gavage with flow slowing down to 60 mL/h) with no modifications in the number and quality of evacuations pro/die. In consideration of the multiple antibiotic therapies delivered during the intensive care stay, a Clostridium Difficile toxin stool test was performed with a negative result. Medication potentially causing diarrhea was suspended (for example, pump inhibitors). Stool chemical physical examination was unremarkable, revealing stools of a pulpy consistency, mustard-colored, with numerous bacteria and rare white blood cells. Coprocolture for *Shigella* and *Salmonella* spp. gave negative results. Blood tests were within normal range except for hypokaliemia. The fecal calprotectin test resulted in normal. Therapeutic fasting of 48 h for the differential diagnosis between malabsorptive and inflammatory was performed, with the persistence of watery diarrhea. Non-specific diarrhea was supposed. An empiric therapy with diosmectite and the probiotic product was started.

After three weeks of intractable watery diarrhea with unchanged clinical and lab examinations, an abdominal CT scan was repeated, showing the absence of the bottom in the stomach with misplacement (see Figure 1B and Figure 3A–C).

A further investigation with abdominal CT enteric contrast media, administered through the feeding tube, was performed. The opacification revealed gastric fundi, transverse, and descending colon with a gastro-colonic communication (Figure 4, Figure 5 and Figure 6).

No pneumoperitoneum, abnormal fluid collections, or peritonitis signs were observed. The diagnosis of gastrocolic fistula was made, and the patient was referred to the surgical department. An attempt to close the fistula by clipping its colic side through a colonoscopy was made. Unfortunately, colic mucosa surrounding the fistula opening resulted in ulcerated and thinner than normal, probably because of persistent gastric acid drainage. The surgical reference center was not much experienced in advanced laparoscopic surgery. A laparotomy was thus planned to perform safe excision of the fistulous tract. Communication between the stomach and colon was identified, and excision of the fistulous tract with primary closure was done without complications. A new PEG was placed uneventfully.

## 3. Discussion

Percutaneous endoscopic gastrostomy is a common procedure used to provide enteral feeding to patients who are unable to take food orally. It involves inserting a feeding tube into the stomach through the abdominal wall using an endoscope. Percutaneous endoscopic gastrostomy is a secure endoscopic technique adopted in many specialized institutions. Complications are uncommon, and among them, the gastrocolonic fistula is one with a minor incidence. It is related to an accidental needle puncture of the colon interposed between the abdominal and gastric walls during the procedure. An asymptomatic period may persist up to several months after the initial PEG placement [6]. Although a gastrocolocutaneous fistula may form at the time of insertion of the PEG tube, symptoms may not manifest until the PEG tube migrates into the transverse colon or until the tube is replaced [7].

When the extremity of the tube migrates into the transverse colon, it becomes clinically evident. Symptoms of gastrocolic fistula can vary depending on the size of the fistula and the amount of material passing through it. Common symptoms include abdominal pain, nausea, vomiting, and diarrhea. Patients may also experience weight loss, electrolyte imbalances, and malnutrition. Sudden onset of diarrhea within minutes after starting enteral feeding, the appearance of fecal material in the PEG tube, or fecaloid vomiting are typical manifestations [8]. Our patient experienced only severe watery diarrhea without any other signs/symptoms. He was a severely impaired brain injury with a disorder of consciousness and was not able to communicate any pain or discomfort. Time-correlation between tube feeding and watery stool emission was missed by the personnel. This could also depend on the patency of the fistula, with discrete residual gastric filling after tube feeding and consequent partial physiological digestive process. Moreover, watery evacuations were also the result of tube hydration, with diarrhea persistence even during the diagnostic fast.

When the migration of a PEG tube into the transverse colon is suspected on clinical grounds, a radiographic study can be performed by administering water-soluble contrast material via the tube to confirm this finding. In addition, an upper endoscopy is recommended for diagnosis (disappearance of gastric botton) and treatment purposes [6].

Data in the previous reports are insufficient to propose standard management for gastrocolocutaneous fistula. Emergency laparotomy as part of the management of these patients is indicated only when peritonitis ensues, as otherwise, spontaneous closure of the colocutaneous and gastrocolic fistulas typically occurs after PEG tube removal [6,9,10]. Liu et al. [9] presented the case of an 87-year-old man with dementia where a percutaneous endoscopic gastrostomy (PEG) tube was placed to treat dysphagia brought on by a sizable, epiphrenic esophageal diverticulum. After PEG feeding for 1 month, he complained of persistent, disabling diarrhea. An extremely unusual side effect of this generally risk-free endoscopic operation is the migration of the PEG tube into the colon, either with or without an associated gastrocolic fistula. A tiny minority (11%) of patients are asymptomatic, with the diagnosis being discovered inadvertently; the majority of patients appear subacutely with debilitating diarrhea (50%) or fecal leakage (39%) several months after first insertion or on tube replacement [6]. This rare condition has also been described by Friedmann et al. [6], who reported six cases of misplacement of the PEG into the colon and reviewed the literature concerning this rare complication. None of them experienced symptoms or signs of peritonitis, and only two patients had a gastrocolonic fistula visible after contrast material was introduced into the tube, likely because it quickly closed. In one of our patients, a surgical gastrostomy was performed, but in the other two, the “pull” procedure to remove the tube and re-insert a new one was successful. Two patients had nasogastric tubes left in place, one due to his poor general health as a result of a separate illness, to which he passed away a month later. Further percutaneous treatment was not possible in the other patient, according to a CT scan, because the bowel was positioned in between the stomach and the abdominal wall. Finally, One of these patients was diagnosed 5 weeks later because of severe diarrhea, hunger, and malnutrition with weight loss. Endoscopic fistula closing can be performed in patients with a large fistula opening that could be not capable of spontaneously close. Even if the fistulous opening is small, endoscopic management will accelerate oral feeding. The majority of previous endoscopic interventions were conducted via colonoscopy by clipping the colonic side of the fistula opening [10].

Of notice, some strategies can be used to prevent accidental misplacement of the PEG tube. First, an elastic bandage can be used to prevent accidental tube traction. Of importance is the length of the tube: the tube needs to be long enough to allow adequate care but not so long to be simply accessible to an agitated patient. Finally, the external bumper needs to be correctly managed to avoid excessive traction or tight skin adherence.

## 4. Conclusions

Migration of a percutaneous endoscopic gastrostomy tube into the transverse colon is often a forgotten cause of refractory diarrhea. Physicians should be aware that misplacement of a PEG tube via a gastrocolic fistula may occasionally occur as a complication of PEG tube placement, often remaining asymptomatic for several months. This condition should be suspected in patients with sudden onset of watery diarrhea after enteral PEG feeding. When clinical suspicion arises, water-soluble contrast material may be administered via the tube to determine if a gastrocolic fistula is present, and a specialistic consultation should be performed.

## Figures and Tables

**Figure 1 healthcare-11-01263-f001:**
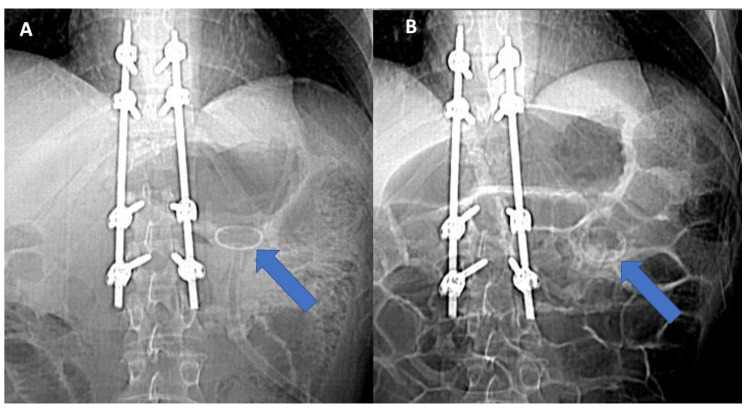
Abdominal AP radiography demonstrating the gastric localization of the PEG botton (arrow) at patient admission time (**A**) and its successive migration in transverse colon lumen (arrow) (**B**).

**Figure 2 healthcare-11-01263-f002:**
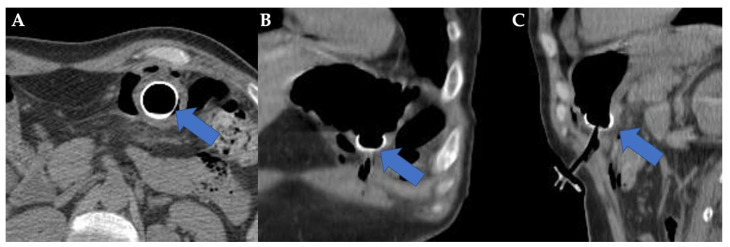
(**A**) Axial, (**B**) Coronal (**C**) Sagittal computed tomography (CT) scans showing the percutaneous endoscopic gastrostomy (PEG) button (arrow) inside the stomach at patient admission time.

**Figure 3 healthcare-11-01263-f003:**
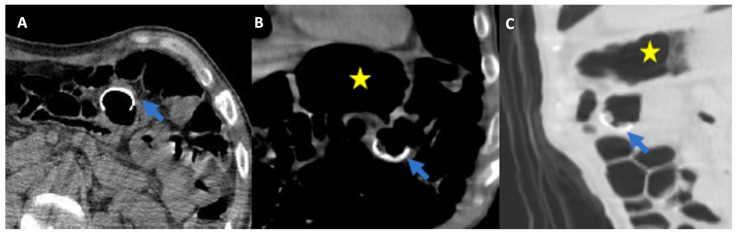
(**A**) Axial, (**B**) Coronal, and (**C**) Sagittal computed tomography (CT) scans showing migration of percutaneous endoscopic gastrostomy (PEG) button in transverse colon lumen (arrow). The gastric lumen is indicated with a yellow star.

**Figure 4 healthcare-11-01263-f004:**
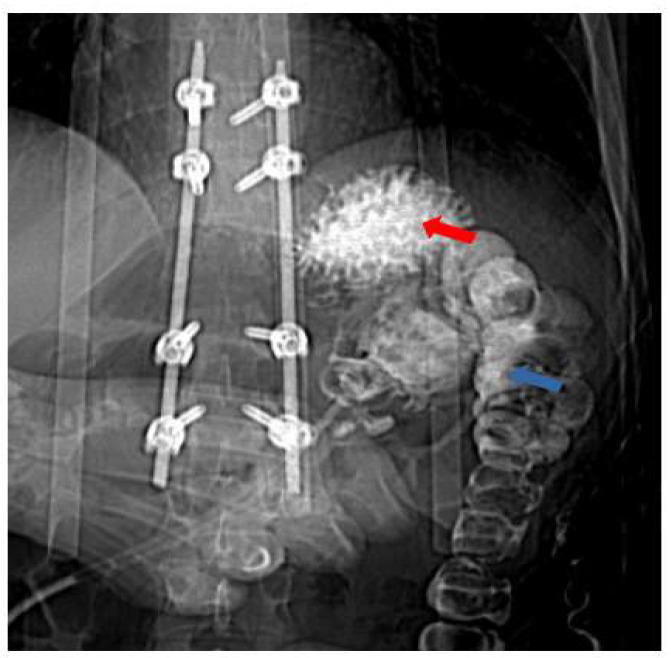
Contrast radiography with gastrografin injected through the gastrostomy tube showing the contrast filling splenic flessure, descending colon (blue arrow), and gastric fundi (red arrow) through the fistula.

**Figure 5 healthcare-11-01263-f005:**
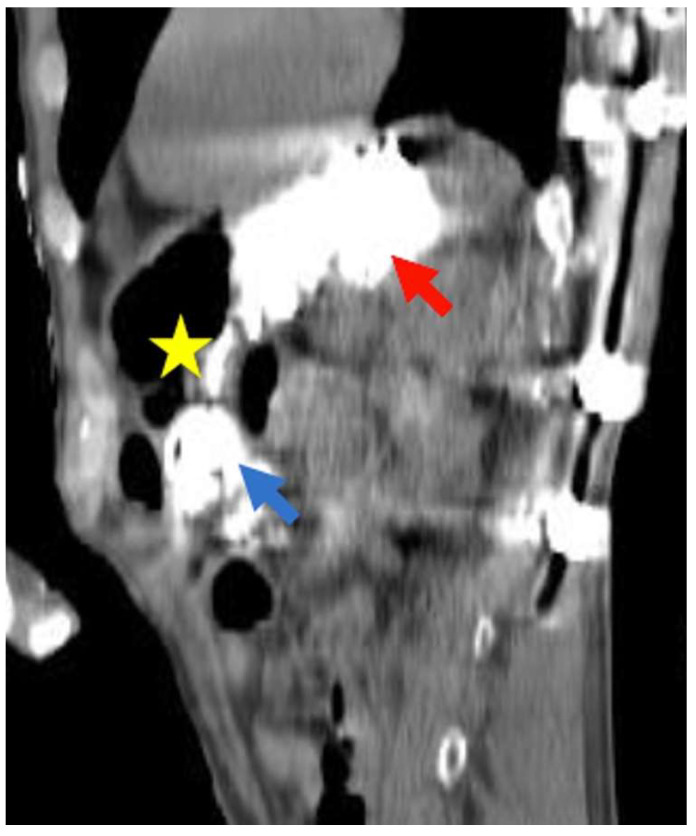
Contrast abdominal CT scan with gastrografin, sagittal view, showing the balloon of the percutaneous endoscopic gastrostomy tube (blue arrow) in the lumen of the transverse colon with contrast filling the stomach (red arrow). The fistula connecting the two structures is clearly visible (yellow star).

**Figure 6 healthcare-11-01263-f006:**
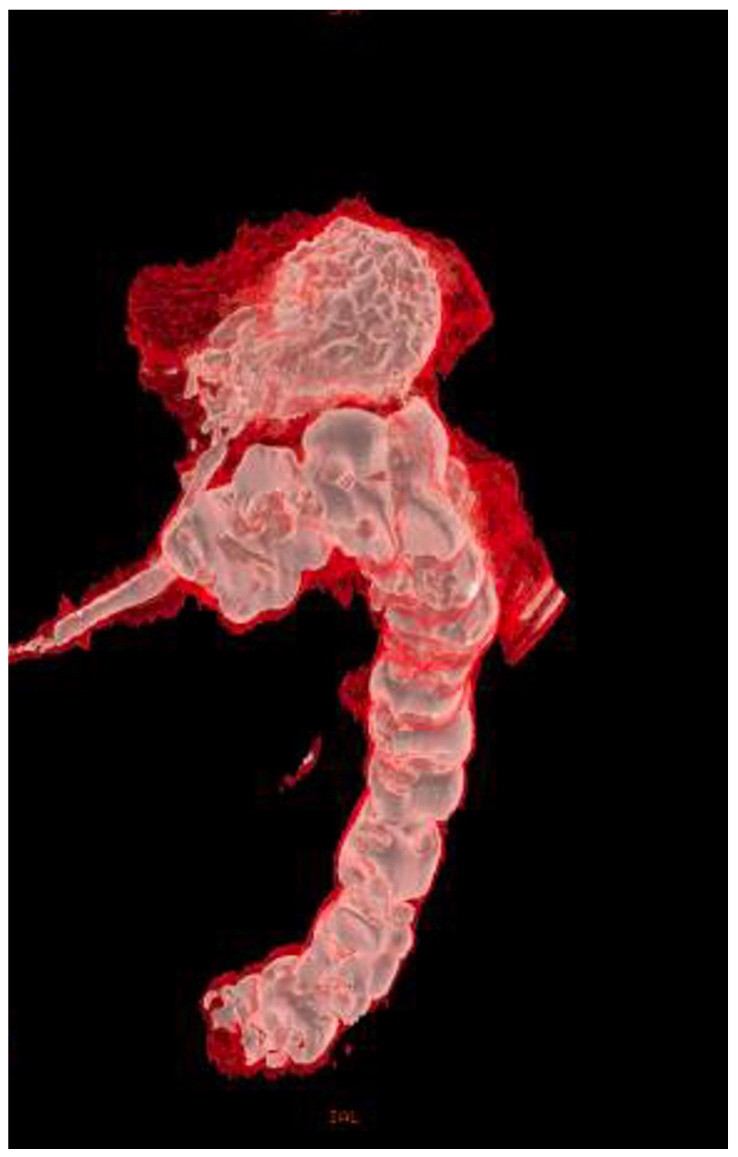
Contrast abdominal CT scan with gastrografin, 3d reconstruction, lateral-oblique view, showing the balloon of the percutaneous endoscopic gastrostomy tube in the lumen of the transverse colon with contrast filling the stomach and descending colon. Again, the fistula connecting the two structures is clearly visible.

## Data Availability

The data presented in this study are available on request from the corresponding author.

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
