# Peer review of "A Rare Case of Severe Diarrhea: Gastrocolic Fistula Caused by Migration of Percutaneous Endoscopic Gastrostomy Tube"

_healthcare, 2023, doi:10.3390/healthcare11091263_

Round 1
Reviewer 1 Report
This case report is clearly described a case of gastrocolic fistula caused by PEG tube. This report is important to alert the clinicians to similar cases in their practice, and there is great potential for this manuscript to be published in “healthcare” as a Case Report. But some revisions of this manuscript are needed before it can be accepted for publication.
Major:
1. The title of the paper should be changed to be more understandable to the reader. For example; “A Rare Case of Severe Diarrhea: Gastrocolic Fistula caused by Migration of Percutaneous Endoscopic Gastrostomy Tube”
2. The authors should describe the strategies to prevent accidental placement of PEG tubes.
3. The authors describe "Emergency laparotomy as part of the management of these patients is indicated only when peritonitis ensues, as otherwise, The authors stated that "Emergency laparotomy as part of the management of these patients is indicated only when peritonitis ensues, as otherwise, spontaneous closure of the colocutaneous and gastrocolic fistulas typically occurs after PEG tube removal". Why was the transverse colon resection performed after removal of the gastrostomy tube even though there were no signs of peritonitis in this case?
Minor:
1. The authors should describe whether there were bloody or tarry stools after PEG placement.
2. The type of PEG used may be clearly stated in the text whether it is a button type or a leak block type.
3. I think that "gastrocolonic fistuale" on the third line of "Discussion" is a misspelling of "gastrocolonic fistula".
Author Response
This case report is clearly described a case of gastrocolic fistula caused by PEG tube. This report is important to alert the clinicians to similar cases in their practice, and there is great potential for this manuscript to be published in “healthcare” as a Case Report. But some revisions of this manuscript are needed before it can be accepted for publication.
Major:
- The title of the paper should be changed to be more understandable to the reader. For example; “A Rare Case of Severe Diarrhea: Gastrocolic Fistula caused by Migration of Percutaneous Endoscopic Gastrostomy Tube”
REPLY: We totally agree that the suggested title is more understandable so we will use it as a new title for the manuscript.
- The authors should describe the strategies to prevent accidental placement of PEG tubes.
REPLY: There are some strategies that can be used to prevent the accidental misplacement of the PEG tube. First, an elastic bandage can be used to prevent accidental tube traction. The tube's length is crucial; it must be long enough to provide sufficient care but not so long that an agitated patient may easily access it. Finally, it's important to properly regulate the external bumper to prevent excessive traction or tight skin adhesion. This additional statement has been included in the discussion
- The authors describe "Emergency laparotomy as part of the management of these patients is indicated only when peritonitis ensues, as otherwise, The authors stated that "Emergency laparotomy as part of the management of these patients is indicated only when peritonitis ensues, as otherwise, spontaneous closure of the colocutaneous and gastrocolic fistulas typically occurs after PEG tube removal". Why was the transverse colon resection performed after the removal of the gastrostomy tube even though there were no signs of peritonitis in this case?
REPLY: We would like to thank this reviewer for this important suggestion. In our case, the surgeon performed an endoscopic exam with the aim of closing the fistula by clipping its colonic side. Unfortunately, the colonic mucosa surrounding the fistula opening resulted ulcerated and thinner than normal, probably because of persistent gastric acid drainage. A laparotomy was thus planned to perform an excision of the fistulous tract. Communication between the stomach and colon was identified, and excision of the fistulous tract with primary closure was done without complications.
Minor:
- The authors should describe whether there were bloody or tarry stools after PEG placement.
REPLY: We added this information.
- The type of PEG used may be clearly stated in the text whether it is a button type or a leak block type.
REPLY: It was a button type PEG.
- I think that "gastrocolonic fistuale" on the third line of "Discussion" is a misspelling of "gastrocolonic fistula".
REPLY: done.
Reviewer 2 Report
In my opinion more surgical details are needed in the report: why did the patient undergo a surgical transverse colic resection if you state clearly that gastrocolic fistula will close spontaneously within days After peg removal?
why laparotomia approach and non a laparoscopic one? Why not performing a feeding jejunostomy since surgery was already required?
Author Response
In my opinion more surgical details are needed in the report: why did the patient undergo a surgical transverse colic resection if you state clearly that gastrocolic fistula will close spontaneously within days After peg removal? why laparotomia approach and non a laparoscopic one? Why not performing a feeding jejunostomy since surgery was already required?
REPLY: We would like to thank this reviewer for this important suggestion. In our case, the fistula was clearly visible in all radiological exams performed, indicating its patency and large caliber. The surgeon performed an endoscopic exam with the aim of closing the fistula by clipping its colic side. Unfortunately, the colonic mucosa surrounding the fistula opening resulted ulcerated and thinner than normal, probably because of persistent gastric acid drainage. A laparotomy was thus planned to perform an excision of the fistulous tract. Communication between the stomach and colon was identified, and excision of the fistulous tract with primary closure was done without complications.
Round 2
Reviewer 1 Report
This manuscript has been revised and has reached a quality worthy of publication in "healthcare". I feel it is accepted in its present form.
Reviewer 2 Report
No further comments